# The Primary Care of Immigrant Workers and Their Associated Characteristics within A Taiwanese Fishing Community

**DOI:** 10.3390/ijerph16193702

**Published:** 2019-10-01

**Authors:** Shih-Chao Kang, Chun-Chi Lin, Chia-Chen Tsai, Yin-Chieh Chang, Chi-Yi Wu, Ke-Chang Chang, Su-Shun Lo

**Affiliations:** 1Daxi Clinic, Toucheng, Yilan, Taiwan; doc1554j@gmail.com (C.-C.L.); jjthsai@yahoo.com.tw (C.-C.T.); 2Division of Family Medicine; National Yang-Ming University Hospital, Yilan 26082, Taiwan; in_J@hotmail.com (Y.-C.C.); wumeowmeow211@gmail.com (C.-Y.W.); 3Faculty of Medicine, School of Medicine, National Yang-Ming University, Taipei 11221, Taiwan; sslo@ymuh.ym.edu.tw; 4Division of Occupational Medicine, National Yang-Ming University Hospital, Yilan 26082, Taiwan; 5Center of Health Management, National Yang-Ming University Hospital, Yilan 26082, Taiwan; 6Department of Otolaryngology, National Yang-Ming University Hospital, Yilan 26082, Taiwan; changandtung@gmail.com; 7Department of Surgery, National Yang-Ming University Hospital, Yilan 26082, Taiwan

**Keywords:** community medicine, fisheries, immigration, labors, primary health care, Taiwan

## Abstract

In Taiwan, immigrant workers play an important role in fisheries but they are easily ignored by society. The health problems and associated characteristics of immigrant workers in fisheries remain unclear. Descriptive and retrospective analyses were performed. Outpatient data were collected from a primary care clinic for six fishing villages in North Eastern Taiwan between 1 August 2016 and 31 July 2017. The data of immigrant workers was recorded and compared with that of natives. A total of 241 immigrant workers and 1342 natives were enrolled. Compared with the natives, the immigrant workers had a significantly younger age, male predominance, and fewer mean visits per year. The immigrant worker’s visits tended to be more highly focused during the third quarter of the year. Immigrant workers paid more registration fees and self-payment, but they paid less on diagnosis fees, oral medication, laboratory exams and had reduced total costs. The top five diagnoses for immigrant workers were respiratory diseases (38.3%), trauma (15.2%), musculoskeletal diseases (11.2%), skin-related diseases (9.5%), and digestive diseases (9.1%). Immigrant workers were positively correlated with infectious/parasitic diseases, and negatively correlated with medical consults and endocrine/metabolic diseases. Immigrant workers were also positively associated with registration fees and self-payment, but negatively correlated with diagnosis fees and total costs (all *p* < 0.05). The distribution of skin diseases and trauma were affected by age and sex as opposed to ethnic group. Immigrant status’ health issues should be given more attention.

## 1. Introduction

Among the primary sector of the economy, the occupational health of fishery workers is noticeable due to their difficult working environments, which have a number of hazards. In the major fishing countries of the West, especially across Northern Europe and Mediterranean nations, fishermen’s health problems and methods of occupational protection are widely discussed. Concerning issues include injuries, infectious diseases, sensory impairments, and skin diseases [1,2,3,4,5,6].

In Taiwan, young populations from rural areas have migrated to metropolitan areas and favored tertiary job sectors that have increased economic development. The provision of labor for primary sectors, including fisheries, has been greatly reduced nationwide. Since 1992, the government of Taiwan has allowed immigrant workers into the country with three-year short-term working visas. In 2018, there were over 670,000 official immigrant workers in Taiwan, including 12,305 working for commercial fisheries [7]. Individuals working in fisheries make up a small proportion of all immigrant workers and their issues may be easily ignored and can be difficult to investigate due to relatively closed and offshore labor environments. The poor conditions of immigrant workers, including their labor rights and environments, have been a global issue, and Taiwan is no exception [8,9]. The roles these individuals have played in commercial fisheries have been invaluable for Taiwan, however the health problems and related characteristics of immigrant workers in fisheries remain unclear. The aim of the current study was to understand the aforementioned issues and make comparisons with native Taiwanese individuals.

## 2. Materials and Methods

### 2.1. Study Design and Sampling

Descriptive and retrospective analyses were performed. Outpatient data were collected from Daxi Clinic, which served as a healthcare center on a remote area of the North-Eastern coast of Taiwan. The community clinic provided primary care for six surrounding fishing villages and was centered within a major fishing port. Except for children and retired elderly, most of the fishing community residents worked as offshore fishermen or onshore fishing processing industry employees. It also provided accessible primary care for immigrant workers. The patients’ demographic data, medical costs, the season of visit, and major diagnosis (using The Tenth Revision of the International Statistical Classification of Diseases and Related Health Problems (ICD-10) codes) were compared between immigrant workers and native patients. The study period was from 1 August 2016 to 31 July 2017.

The items of medical costs were collected from in-clinic statistical claim data for National Health Insurance (NHI) payments. For the description of medical costs, ‘self-payment’ means 1) the medical costs paid by those who did not join the National Health Insurance of Taiwan (NHI), 2) services that the NHI did not pay for, such as intravenous/intramuscular injections of painkillers, nutrients or antibiotics. Those who received a routine NHI-regulated health survey, cancer screening services, or had incomplete data were excluded from the study. The false hypothesis of the present study was that immigrant workers had similar health problems to natives within fishing communities due to high homogeneity of working styles. The current study was approved by the Institutional Review Board of National Yang-Ming University Hospital (approval no. 2019A016) which included ethical approval. All the subjects were anonymous for protection.

### 2.2. Statistical Analysis

The Health Information System for Clinics (Vision Asia Tech. Ltd., Taipei, Taiwan) was mined for data. Data were expressed as the mean ± standard deviation (SD) or percentage (%). All statistical analyses were performed using SPSS software (IBM SPSS version 22.0, SPSS Inc., Chicago, Illinois, USA). An independent *t*-test, chi-square test, Fishers’ exact test, Pearson chi-square test, and multivariate logistic regression analysis were performed. The correlated factors were adjusted by age and sex and presented as hazard ratios (HRs). A *p*-value < 0.05 was considered to be statistically significant.

## 3. Results

### 3.1. Demographics

During the research period, a total of 1583 subjects who made 9090 visits, were enrolled from the fishing community-based clinic, this included 241 immigrant workers and 1342 natives. Among the immigrant workers, 195 were originally from Southeast Asia (including Indonesians, Vietnamese, and Filipinos) and had official permits covered by the NHI. The other 46 were originally from the People’s Republic of China (PRC) and they were non-officially introduced to the country and were not covered by the NHI. The demographic data of all study participants are listed in Table 1. Compared with the natives, the immigrant workers had a significantly younger mean age, a higher proportion of males, and fewer mean clinic visits per year. All the immigrant workers were aged <65 years old. 

### 3.2. Seasonal Effects and Medical Costs

The quarters of a year were divided per three months. For instance, the first quarter includes January to March, the second quarter April to June, the third quarter July to September, and the fourth quarter October to December. Immigrant workers tended to visit more frequently in the third quarter of the year, while natives visited equally throughout all seasons. Immigrant workers paid more registration fees and had higher self-payments, but they paid lower fees for diagnosis, oral medication, and laboratory exams, and had reduced total costs compared with the natives (Table 2).

### 3.3. Distribution of Major Diagnoses

Using ICD-10 codes, the major diagnosis for immigrant workers differed significantly from the natives, and their top five diagnoses were respiratory diseases (ICD J00-J99, 38.3%), trauma (ICD S00-T98, 15.2%), musculoskeletal diseases (ICD M00-M99, 11.2%), skin-related diseases (ICD L00-L99, 9.5%), and digestive diseases (ICD K00-K93, 9.1%). Skin-related diseases and trauma were diagnosed in a notably higher proportion of immigrant workers compared with the natives (Table 3). 

### 3.4. Multi-Variate Logistic Regression Analyses

Table 4 shows the significance of various factors correlated with immigrant workers, compared with age and sex-adjusted natives. Multivariate logistic regression revealed that immigrant status was significantly associated with visits in the first quarter of the year compared with the fourth quarter of the year. Immigrant workers were also positively associated with infectious diseases (A00-B99), and negatively associated with medical consults (Z00-Z99) and endocrine/metabolic diseases (E00-E90). Immigrant workers were positively correlated with registration fees and self-payments, but negatively associated with diagnosis fees and total cost (all *p* < 0.05).

## 4. Discussion

In Taiwan, there have been several previous studies regarding medical care for immigrant workers and new residents (immigrants by marriage) since the introduction of policy for receiving immigrant workers in 1992. However, most of the local studies have focused on the health of immigrant women and their children, such as peripartum/postpartum care, oral health, and transcultural adaptation and psychological illness [10,11,12,13]. Studies into immigrant fishermen’s health and their medical care remain scarce and are universally ignored by the media. At present, there is only one association focusing on the labor rights and medical care of immigrant fishermen [14]. By evaluating the use of primary care, the current study aimed to understand the health problems of this unique group of immigrant workers, and to discuss possible improvements.

The demographic data revealed that the majority of immigrant workers are male. Under the traditional gender divisions of labor in fishing villages, male immigrant workers serve as fishermen, while the females remain home as family caretakers or employees of peripheral on-shore fishing processing industries. Immigrant workers had fewer clinic visits compared with the natives, which may be due to their lower age as well as the language-related medical barrier. When immigrant workers from Southeast Asia, the majority of which do not speak fluent Mandarin, were compared with those from PRC, who speak Mandarin and are able to easily communicate with natives, immigrant workers from Southeast Asia had notably fewer clinic visits (1.92 ± 1.72 vs. 2.00 ± 1.33; p = 0.76). The authors assumed that such barriers were due to short-term working visas which rotated per three years and scarce pre-occupational language training. Such barriers also affected the depth of clinical interview and major diagnosis which made it difficult to approach psychological issues for immigrant workers. A previous local study of immigrant women/new residents’ health also revealed that language barriers created problems [8]. The authors think that real-time translation assistants and basic Indonesian or Vietnamese medical language training for medical crew could improve immigrant workers’ health care.

The seasonal distribution of visits revealed certain characteristics of immigrant workers in fisheries. The predominance of visits during the third quarter of the year for immigrant workers, reflects the homogeneity of their life and work. The majority of study participants were fishermen and their increase in visits was associated with the major fishing period which starts before the Mid-Moon Festival (around August to September). In contrast, the natives, which included more elderly, adolescents, and children, had more heterogeneity in their life and work. Therefore, the seasons did not have such a large effect on their visits.

The medical costs also reflected some issues. Natives paid lower registration fees compared with immigrant workers because natives had NHI-regulated copayment exempting identifications, such as for the elderly, handicapped, veterans, children under three years old and low-income families [15]. Immigrant workers paid fewer fees for diagnosis, oral medications, and laboratory exams. These results indicate that their visits focused on acute illness and short-term prescriptions, and this is reflected in the lower total costs observed. Interestingly, immigrant workers paid more self-payments than natives. After excluding those from PRC who were not covered by NHI, immigrant workers from Southeast Asia still paid significantly more self-payments than natives (75.51 ± 99.43 vs. 38.77 ± 75.73; *p* < 0.001). This phenomenon reflects the fact that some therapies and medications are not paid for by NHI, such as intramuscular or intravenous injections of painkillers, nutrients or antibiotics, which were commonly required by the immigrant workers.

The distribution of major diagnoses highlighted some issues. Based on the top five major diagnoses, respiratory, musculoskeletal, and digestive diseases were common in both groups. However, trauma and skin-related diseases were more dominant in immigrant workers. After age-sex adjustment, the correlation between immigrant workers and skin diseases/trauma became insignificant. These results reflect the similarities in the working- and life-styles of the immigrant workers and age-sex matched natives. Many from each group worked as fishermen, and therefore, suffered from similar health problems, which were related to their occupation. This also explains why the correlation between the third quarter of the year and immigrant workers became insignificant after age-sex matched adjustment was performed. These types of illness are commonly observed in fishermen and they have been previously reported. A Moroccan study highlighted that a low understanding of occupational safety and the limited use of protective equipment were risk factors for skin diseases [16]. Even in the United States, compliance with safety regulations and fishermen’s safety training were not commonly implemented in commercial fisheries [17]. In Denmark, it has been reported that fishermen’s musculoskeletal diseases were associated with their work type and load, such as long-term standing, repetitive hand and finger movement, and twisting and bending of the back [18]. These results reflect the importance of occupational safety and working conditions in fisheries. More attention should be paid to improving these conditions.

The multivariate logistic regression revealed the co-relationship between certain variables and immigrants in contrast to age-sex matched natives. The positive correlation between the first quarter and immigrant workers’ clinic visits reflects the seasonal effect on this patient group. In contrast to age-sex matched natives, most of the immigrant workers come from tropical regions and cold intolerance is common, which can affect their working and health status. The high prevalence and high HR of infectious/parasitic diseases in immigrant workers also reflect some health problems associated with tropical medicine and environmental hygiene. The negative correlation between health consults (ICD Z00-Z99) and endocrine diseases (ICD E00-E90) in immigrant workers supports two major notions: 1) The majority of their clinic visits were for acute health problems as opposed to consults, 2) they had been artificially screened before being introduced to Taiwan and those with chronic diseases were previously excluded.

Compared with age- and sex-matched natives, the positive association between the registration fee and self-payment with immigrant workers revealed that they did not share NHI-regulated copayment exemptions for special groups, and were required to pay for significantly more therapies, as previously described (Table 2). It is well known that intramuscular or intravenous injections can transiently relieve symptoms but this has little effect on the whole course of treatment. For immigrant workers who rely on physical laboring, transient relief is welcome, but it is purely symptomatic. This indicates that they are being pushed to work as hard as possible by their employers. It tells us that labor environments under socioeconomic pressure could affect a patient’s medical behavior and such influences would not be beneficial for to them. The overuse of injections in general practice has been problematic in other Asian countries, such as the PRC, Pakistan, Cambodia, Nepal, India, and South Korea, and the introduction of national policies presented limited improvement [19,20,21,22,23,24,25,26]. It should be noted that improvements to immigrant workers’ health and their right to rest and medical attention, as well as an increase in health education for employers and their employees, are both urgently required. 

### Limitations

There were several limitations to the current study. The claim data itself were designed for NHI payments, and their statistical variables were limited. This study could only describe the situation of official immigrant workers and not of those with unofficial/illegal status. Subjects’ comorbidities were not included due to limitations of the claim data. Some items, which could have been listed on the demographic data, such as occupation, marital status, and history of substance use (cigarette or alcohol), were lacking due to incomplete chart records. The medical services and diagnostic tools of the primary care clinic were limited and this may have affected the accuracy of the major diagnoses, especially those listed as ICD R00-R99.

## 5. Conclusions

In fishing communities, immigrant workers had distinct demographic characteristics compared with the native population, such as younger age, male predominance, and heterogeneity of visiting seasons. The distribution of skin diseases and trauma were affected by age and sex. Compared with age- and sex-matched natives, the immigrant workers had higher HRs for infectious/parasitic diseases, and lower ratios for emergency medical consults and endocrine/metabolic diseases. Immigrant workers favored self-payment of medical services, such as intravenous/intramuscular injections, which reflects the reason for their medical therapies and an underlying labor environment under socioeconomic pressure. Based on our analyses, we believed that more attention should be given to healthcare status and labor environments of immigrant fishery workers in Taiwan, and improvements should be implemented.

## Figures and Tables

**Table 1 ijerph-16-03702-t001:** Data on subjects (*n* = 1583).

Items	Immigrant workers *(*n* = 241)	Natives (*n* = 1342)	*p*
Mean age (y/o)	36.23 ± 9.91	47.21 ± 23.29	<0.001
Sex (male/female)	210/31	694/648	<0.001
Age ≥65 years old (yes/no)	0/241	316/1026	<0.001
Mean visits per year (times)	1.95 ± 1.67	6.42 ± 9.37	<0.001

* Including 195 subjects from Southeast Asia and 46 subjects from the People’s Republic of China.

**Table 2 ijerph-16-03702-t002:** Season of visit and cost of the subjects’ visit (*n* = 9090).

Parameters	Immigrant Workers (*n* = 473)	Natives (*n* = 8617)	*p*-Value
**Season of visits**			0.044
Quaternary 1	117	1985	
Quaternary 2	120	2204	
Quaternary 3	140	2221	
Quaternary 4	96	2207	
**Costs (TWD *)**			
Fee of registration	44.50 ± 26.14	33.39 ± 27.10	<0.001
Fee of diagnosis	187.90 ± 112.51	232.82 ± 128.68	<0001
Fee of oral medications	60.40 ± 83.95	155.10 ± 305.17	<0.001
Fee of topical medications	5.07 ± 16.88	5.78 ± 22.58	0.502
Fee of laboratory exams	6.72 ± 67.51	40.09 ± 175.42	<0.001
Fee of self-payment	128.03 ± 149.47	38.78 ± 75.25	<0.001
Fee of total costs	416.29 ± 142.41	494.24 ± 393.40	<0.001

* TWD, New Taiwan Dollars.

**Table 3 ijerph-16-03702-t003:** List of major diagnosis of subjects’ visits *.

Classification of ICD-10 codes (%)	Immigrant Workers (*n* = 473)	Natives (*n* = 8617)
A00-B99 Infectious and parasite diseases	18 (3.8)	88 (1.0)
C00-D48 Neoplasms	0 (0)	2 (0.02)
D50-D89 Diseases of the blood and blood-forming organs and certain disorders involving the immune mechanism	0 (0)	34 (0.4)
E00-E90 Endocrine, nutritional and metabolic diseases	1 (0.2)	560 (6.5)
F00-F99 Mental and behavioral disorders	0 (0)	174 (2.0)
G00-G99 Diseases of the nervous system	2 (0.4)	125 (1.5)
H00-H59 Diseases of the eye and adnexa	2 (0.4)	49 (0.6)
H60-H95 Diseases of the ear and mastoid process	6 (1.3)	164 (1.9)
I00-I99 Diseases of the circulatory system	14 (3.0)	1147 (13.3)
J00-J99 Diseases of the respiratory system	181 (38.3)	2586 (30.0)
K00-K93 Diseases of the digestive system	43 (9.1)	684 (7.9)
L00-L99 Diseases of the skin and subcutaneous tissue	45 (9.5)	499 (5.8)
M00-M99 Diseases of the musculoskeletal system and connective tissue	53 (11.2)	1157 (13.4)
N00-N99 Diseases of the genitourinary system	8 (1.7)	204 (2.4)
O00-O99 Pregnancy, childbirth and the puerperium	0 (0)	2 (0.02)
R00-R99 Symptoms, signs and abnormal clinical and laboratory findings, not elsewhere classified	26 (5.5)	528 (6.1)
S00-T98 Injury, poisoning and certain other consequences of external causes	72 (15.2)	526 (6.1)
V01-Y98 External causes of morbidity and mortality	0 (0)	4(0.04)
Z00-Z99 Factors influencing health status and contact with health services	2 (0.4)	85 (1.0)

* *p* < 0.001 by Pearson Chi-square test.

**Table 4 ijerph-16-03702-t004:** Correlation factors of immigrant workers compared with natives following multivariate regression analyses.

Factors	Adjusted ^a^ HR	*p*-Value
**Season of visits**		
Quaternary 1	1.411	0.019
Quaternary 2	1.101	0.546
Quaternary 3	1.220	0.195
Quaternary 4	-	-
**Major diagnosis, ICD-10 codes**		
A00-B99 Infectious and parasite diseases	3.196	0.014
J00-J99 Diseases of the respiratory system	0.637	0.206
K00-K93 Diseases of the digestive system	0.712	0.388
L00-L99 Diseases of the skin and subcutaneous tissue	1.314	0.492
M00-M99 Diseases of the musculoskeletal system and connective tissue	0.787	0.521
N00-N99 Diseases of the genitourinary system	0.959	0.937
O00-O99 Pregnancy, childbirth and the puerperium	< 0.001	0.999
R00-R99 Symptoms, signs and abnormal clinical and laboratory findings, not elsewhere classified	0.874	0.747
S00-T98 Injury, poisoning and certain other consequences of external causes	1.922	0.097
C00-D48 Neoplasms	< 0.001	0.999
V01-Y98 External causes of morbidity and mortality	< 0.001	0.999
Z00-Z99 Factors influencing health status and contact with health services	0.085	0.039
D50-D89 Diseases of the blood and blood-forming organs and certain disorders involving the immune mechanism	< 0.001	0.998
E00-E90 Endocrine, nutritional and metabolic diseases	0.088	0.022
F00-F99 Mental and behavioral disorders	< 0.001	0.995
G00-G99 Diseases of the nervous system	0.515	0.451
H00-H59 Diseases of the eye and adnexa	1.141	0.876
H60-H95 Diseases of the ear and mastoid process	0.629	0.437
I00-I99 Diseases of the circulatory system	-	-
Fee of registration	1.065	< 0.001
Fee of diagnosis	0.989	< 0.001
Fee of oral medications	1.002	0.286
Fee of topical medications	1.001	0.762
Fee of laboratory exams	1.002	0.362
Fee of self-payment	1.012	< 0.001
Fee of total costs	0.995	0.017

^a^ After adjustments for age and sex. Abbreviations: HR, hazard ratio.

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
