# Peer review of "The Primary Care of Immigrant Workers and Their Associated Characteristics within A Taiwanese Fishing Community"

_ijerph, 2019, doi:10.3390/ijerph16193702_

Round 1
Reviewer 1 Report
Indeed, this research is important in shedding light on the health issues experienced by immigrants working in fisheries in Taiwan. However, there are several items the authors still need to revisit and address in their manuscript. They are as follows:
There are several writing mechanics issues throughout the paper (i.e., grammar and other copy edit issues). For example in the abstract, the last sentence (line 40) "...should be paid more attention and improvements" needs to be revised (i.e., improvements in what?). Another example is misspelling in line 91--"Philippine" should be spelled as "Philippines". There are more in other parts of the manuscript, and the authors should consider doing another round of copy editing before resubmission. The INTRODUCTION section lacks a thorough understanding of the literature in commercial fishing occupation safety and health, especially among immigrant populations. For example, the authors did not mention specific health issues specifically experienced by immigrants working in commercial fishing and/or other related work. The literature on immigrants in fishing and agricultural work suggest that they have poorer health outcomes, and they are subject to poorer work conditions. Thus, I don't follow why the authors made the hypothesis that immigrants have "similar" health problems as their non-immigrant counterparts (by the way, I would rather see the authors' hypothesis in the Introduction section as opposed to the Methods section). Also in this section, the authors do not expand on their claim that fisheries have "unfriendly labor environments" (line 55) In the METHODS section, the authors did not specify whether their study sample comes from all sectors of the fishing industry. In other words, does their sample include both fishermen and fish processing workers? It's also not clear how many different companies are in the fishing villages. And does the study sample include workers onshore and offshore? The Methods also does not discuss how cost was measured. In the RESULTS section, the authors mentioned 241 workers, but they did not provide the proportion of where the immigrants are coming from. It's also not clear how the quarters are divided in the fishing industry in Taiwan. It is also not clear in this section why age was dichotomized into < 65 years and >/= 65 years. In the DISCUSSION section, the authors addressed their hypothesis related to health problems. However, they did not address their hypothesis related to "attending behaviors". Perhaps, I'm not clear what "attending behaviors" are as introduced in the Methods section. In the CONCLUSION, the authors mention language barriers among immigrants, but there was no data or literature to support this claim (in other words, are immigrant workers hired even without knowing the Taiwanese language?). Lower socioeconomic status was also mentioned here regarding immigrant workers; but again, there's no evidence or literature to support this claim. If they are from low socioeconomic status, how are they able to afford going to Taiwan? Or are the fishing companies providing all the travel cost?
Author Response
Dear Reviewer 1.
Thank you for your quotable comments. The replications for your comments were listed as below.
1. The manuscript had been English-edited by ATS Medical Editing (see the attached certificate). The spelling was also corrected and unified.
2. For Introduction, we had added the sentences:” The poor conditions of immigrant workers for their labor rights and environments had been a global issue, and Taiwan could not be excepted.”
3. For Materials and Methods, the hypothesis of this study was false hypothesis (H0) and the sentences had been corrected as “The false hypothesis of the present study was that immigrant workers had similar health problems to natives within fishing communities due to high homogeneity of working styles.”
The norm “unfriendly labor environments” was not clear and had been shifted as “illogical labor environments” and unified.
The characteristics of setting fishing communities and samples had been explained as sentences,” Except children and retired elderly, most of the fishing community residents worked as offshore fishermen or onshore fishing processing industry employees.”
The medical costs were calculated by automatic in-clinic claim systems and described as “The items of medical costs were collected from in-clinic statistical claim data for National Health Insurance (NHI) payments.”
4. For Results, the nationalities of immigrant workers were decided by official policies and diplomatic strategies. For immigrant fishermen, Indonesians accounted mostly and Vietnamese were introduced secondary. Filipinos accounted minimal due to diplomatic conflicts between Taiwan and Philippines. The confirmation of their identifications in our clinic relied on NHI cards which marked Roman-spelling names instead of nationalities. Vietnamese were easily distinguished due to their unique spelling names, but it was difficult to distinguish between Indonesians and Filipinos. This is, we included them as South East Asia for convenience of research.
The quarters of a year had been explained in Result 3.2.
5. For Discussion, the norm “attending behaviors” was not adequate for this study. We had deleted the norm and unified.
6. For conclusion, the relevant sentences about language barriers and low socio-economic status of immigrant workers had been deleted.
Thank you very much. Please contact me as soon as possible.
Sincerely,
Shih-Chao Kang

Reviewer 2 Report
Well researched paper. Sound methodology, the findings are supported by the background research. English should be checked and improved.
Author Response
Dear Reviewer 2.
Thank you for your encouragements. The replications for your comments were listed as below.
1. The manuscript had been English-edited by ATS Medical Editing (see the attached certificate). The spelling was also corrected and unified.
Thank you very much. Please contact me as soon as possible.
Sincerely,
Shih-Chao Kang

Reviewer 3 Report
This is an interesting topic and needs to profiled more in the research literature. However, the paper needs several improvements and I would be hesitant to recommend it for publication in its current form. Language needs some improvement Immigrant worker as a concept needs to be defined in the Taiwanese context I am not sure if the study is about behaviours per se. It is some sort of a pattern of health care consumption which seems to be influenced by working conditions. The information on the database utilized seems limited. The quality control data, how the data are registered needs to be explained. The different types of fees that are used the analysis need to be explained in detail. There was little information if the diseases and injuries were work-related or not Also, hardly any information on psychosocial problems among the migrant workers Under-reporting of diseases and injuries among migrants has been documented globally. This aspect needs to be addressed in the discussion. The lack of information on how long the immigrants have stayed in the country, occupation (job-tasks), socioeconomic status are major limitations in this study. These data are potential confounders and some discussion on these limitation seems legitimate. Conclusions on language issues and socioeconomic status are pertinent, but not seem to be supported by the data provided.
Author Response
Dear Reviewer 3.
Thank you for your quotable comments. The replications for your comments were listed as below.
The manuscript had been English-edited by ATS Medical Editing (see the attached certificate). The spelling was also corrected and unified. The norm “attending behaviors” was not adequate for this study. We had deleted the norm and unified. The description of dataset limitations had been added in Limitations as” The claim data itself was designed for NHI payments, and its statistical variables were limited.” The explanation of quality control of data was seen in Materials and Methods,” Those who received a routine NHI-regulated heath survey, cancer screening services, or had incomplete data were excluded from the study.” The scarce of psychological problems in immigrant workers was supposed by language barriers. The relevant sentences had added as” The authors supposed that such barriers were due to short-term working visas which rotated per three years and scarce of pre-occupational langrage training. Such barriers also affected the depth of clinical interview and major diagnosis which made it difficult to approach psychological issues for immigrant workers.” (see Discussion) We had added in Introduction as “The poor conditions of immigrant workers for their labor rights and environments had been a global issue, and Taiwan could not be excepted.” We had deleted the description of low socio-economic status and language barriers in Conclusion due to scarce of evidence in Results. The short-term working visas for immigrant workers in Taiwan were per three years and described in Introduction and Discussion.
Thank you very much. Please contact me as soon as possible.
Sincerely,
Shih-Chao Kang

Round 2
Reviewer 1 Report
The authors have addressed most of my concerns. One area that can be improved is in the Introduction section (line 57-59), the authors did not provide evidence or citation that "poor conditions of immigrant workers...had been a global issue." And perhaps the author can describe why that is.
Author Response
Dear Reviewer 1,
As your comment, I have added citation [8,9] on line 57. Thank yu very much.
Sincerely,
Shih-Chao Kang
